# EFFECTIVE PRUNING OF WEB-SCALE DATASETS BASED ON COMPLEXITY OF CONCEPT CLUSTERS

**Amro Abbas**[*†], **Evgenia Rusak**[1*†], **Kushal Tirumala**[2], **Wieland Brendel**[3,4,5],
**Kamalika Chaudhuri**[2,6], **Ari S. Morcos**[7‡]

`amrokamal30@gmail.com`, `evgenia.rusak@uni-tuebingen.de`
University of Tübingen, Germany[1]     Meta AI (FAIR)[2]
ELLIS Institute Tübingen[3] Max-Planck Institute for Intelligent Systems[4] Tübingen AI Center[5]
University of California San Diego[6]     DatologyAI[7]

## ABSTRACT

Utilizing massive web-scale datasets has led to unprecedented performance gains in machine learning models, but also imposes outlandish compute requirements for their training. In order to improve training and data efficiency, we here push the limits of pruning large-scale multimodal datasets for training CLIP-style models. Today's most effective pruning method on ImageNet clusters data samples into separate concepts according to their embedding and prunes away the most prototypical samples. We scale this approach to LAION and improve it by noting that the pruning rate should be concept-specific and adapted to the complexity of the concept. Using a simple and intuitive complexity measure, we are able to reduce the training cost to a quarter of regular training. By filtering from the LAION dataset, we find that training on a smaller set of high-quality data can lead to higher performance with significantly lower training costs. More specifically, we are able to outperform the LAION-trained OpenCLIP-ViT-B/32 model on ImageNet zero-shot accuracy by 1.1p.p. while only using 27.7% of the data and training compute. Despite a strong reduction in training cost, we also see improvements on ImageNet dist. shifts, retrieval tasks and VTAB. On the DataComp Medium benchmark, we achieve a new state-of-the-art ImageNet zero-shot accuracy and a competitive average zero-shot accuracy on 38 evaluation tasks.

## 1 INTRODUCTION

Scaling the model and the training dataset size has been shown to increase performance across a wide range of tasks (Djolonga et al., 2021; Zhai et al., 2022; Kolesnikov et al., 2020; Taori et al., 2020). Foundation Models (Bommasani et al., 2021) such as CLIP (Radford et al., 2021b), DinoV2 (Oquab et al., 2023), LLaMA and LLaMA-2 (Touvron et al., 2023a;b) or Eva Fang et al. (2023) have revolutionized the Deep Learning field and sparked interest beyond the academic realm with their unprecedented capabilities in vision and language. However, training (foundation) models on larger datasets incurs high computational and environmental costs which are out of reach for most academic labs.

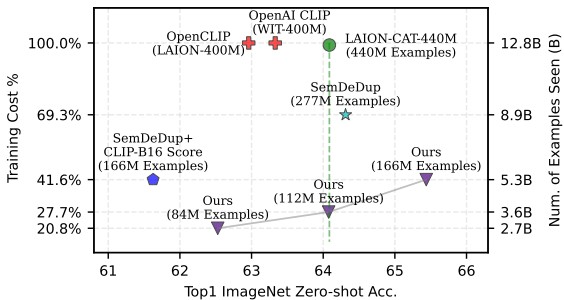

Figure 1: With our approach, we outperform training on the full LAION-400M dataset (64.1% vs 63.0%) for CLIP-ViT-B/32 models while significantly reducing the training cost to 27.7%. We filter from the LAION-CAT-440M by first deduplicating it to 277M examples using the SemDeDup method and then applying Density-Based Pruning (DBP) to get datasets of sizes 84M, 112M, and 166M examples.

---

[*]Equal contribution. [†]Work done during an AI residency (Amro) / research internship (Evgenia) at Meta AI (FAIR). [‡]Work done while at Meta AI (FAIR). Code at `github.com/amro-kamal/effective_pruning`.

In contrast to the highly curated ImageNet dataset (Deng et al., 2009), web-scale datasets such as LAION (Schuhmann et al., 2022) are noisy, and filtering out less informative data can strongly improve data efficiency and speed up learning. For example, discarding images below a certain CLIP-score, which is the cosine similarity between image and caption embeddings, has been shown to improve data efficiency. The original LAION dataset used a CLIP-score value of 0.3 as one of the steps to create LAION-400M (Schuhmann et al., 2021b). In the recently proposed benchmark DataComp (Gadre et al., 2023), which aims to find optimal data for a broad range of downstream tasks, CLIP-score filtering has emerged as a strong baseline (Gadre et al., 2023). Apart from CLIP-score filtering, other works assessed the complexity and the action-content of individual captions, and removed images containing (parts of) the caption via text-spotting (Radenovic et al., 2023), or used the CLIP-score to gauge the importance of the captions within the image before removing them (Maini et al., 2023).

So far, other works on pruning of large-scale datasets have focussed on assessing the quality of individual data samples. We argue that the marginal importance of a data point depends on other data points in its vicinity, such that optimal dataset coverage allows to discard more data points from denser regions, while keeping more data points from sparser regions. In order to achieve this, we begin by scaling the simple and theoretically motivated Self-Supervised-Prototypes Pruning method (SSP-Pruning, Sorscher et al., 2022) to web-scale datasets. To recap SSP-Pruning, Sorscher et al. (2022) proposed to cluster the embeddings of a pretrained model with k-means, then ranked all samples by their distance to the nearest cluster centroid and pruned the dataset by discarding the most prototypical examples. With their approach, Sorscher et al. (2022) outperformed all other pruning methods on ImageNet. Since SSP Pruning has been shown to scale to large language models (Tirumala et al., 2023), we take this method as the most promising technique on ImageNet, and investigate which steps are necessary to scale it to CLIP training on LAION; we also modify the pruning criterion by considering the complexity of concepts in the dataset. On a high level, we wish to have approximately the same sample density across the whole embedding space, thus, we call our method Density-Based-Pruning (DBP).

Our contributions are:

- We scale SSP-Pruning to web-scale datasets which is a non-trivial task involving a deduplication step, investigate how the complexity of different concepts within a dataset can be used for pruning, and report further improvements over regular SSP-Pruning.
- We demonstrate that the pruning criterion we developed on LAION also transfers to the DataComp benchmark (Gadre et al., 2023), and beat the current state of the art reported in the literature in most categories.
- We show empirically that training on smaller high-quality data can result in a better model with significantly lower training cost.

## 2 RELATED WORK

### 2.1 DATA CURATION IN SUPERVISED LEARNING

Our work is related to *coreset selection* which focuses on identifying a highly informative subset of a large dataset to improve training efficiency. Usually, samples which are considered to be harder based on some scoring criterion are kept while easier examples are discarded. Criteria for ranking the samples are based on (dynamic) model uncertainty (Gal et al., 2017; Coleman et al., 2019; He et al., 2023), distance of samples to score medians (Xia et al., 2022), the average L2 norm of the error vector (Paul et al., 2021), the degree of forgetting over the course of training (Toneva et al., 2018), the degree of memorization (Feldman & Zhang, 2020; Feldman, 2020), and many others. Killamsetty et al. (2021) propose an iterative bi-level optimization to select the coreset from a large pool of unlabeled data which would result in minimum labeled set loss when trained upon in a semi-supervised manner.

### 2.2 CONTRASTIVE IMAGE-LANGUAGE PRETRAINING

Combining caption supervision with training of large models on large-scale datasets has transformed the field of computer vision, and models trained with CLIP (Radford et al., 2021b) or ALIGN (Jia et al., 2021) have shown exceptional performance across a range of down-stream tasks, such as image

generation (Ramesh et al., 2022), image segmentation (Xu et al., 2023b), text-to-image synthesis (Li et al., 2022b), video understanding (Xu et al., 2021), and others. The open-source projects OpenCLIP (Ilharco et al., 2021) and LAION-2B (Schuhmann et al., 2022) have democratized research on large-scale multimodal models and have been crucial to make such progress possible. Still, while training of large-scale models on large-scale datasets is possible in theory, it remains prohibitively expensive for most academic labs in practice: For example, training of the ViT-L/14 model (Dosovitskiy et al., 2020) with OpenCLIP took 400 A100 (40 GB) GPUs for around 127 hours.

## 2.3 DATA CURATION AT SCALE

There exist different strategies to make CLIP training more efficient. We split the data curation methods based on the way they filter the data into three categories, although overlaps exist.

**Redundancy Reduction**    This category of methods aims to reduce data redundancy by removing duplicates as in Abbas et al. (2023); Webster et al. (2023). These methods consider the similarity between examples in the data population and remove samples whose similarity falls below a pre-defined threshold. This results in more balanced data and saves training costs spent on training on semantically similar examples.

**Matching Score Filtering**    This category of methods ranks the individual examples using an Image-Text matching (ITM) score computed using a pre-trained model like CLIP (Radford et al., 2021b) or BLIP (Li et al., 2022a). A simple and strong baseline for ITM filtering is the CLIP-score which is the cosine similarity of image and text token embeddings of a pretrained CLIP model. The LAION-400M dataset itself has been filtered using the CLIP-score such that image-caption pairs were discarded if their CLIP-score was below 0.3 (Schuhmann et al., 2021b). The CLIP-score is also a strong baseline on subsets of all scales in the DataComp benchmark (Gadre et al., 2023).

**Improving the data quality**    It has been shown that data efficiency can be improved by diversifying (Santurkar et al., 2022) or denoising the captions (Nguyen et al., 2023) or by using shorter image/ text token sequences for larger image/ text encoders during CLIP training (Li et al., 2023). Xu et al. (2023a) incorporates a data objective into their training pipeline and dynamically selects data during training. Radenovic et al. (2023) remove examples with short captions and examples with low caption complexity. In addition, they remove examples that contain part of the caption as text in the image to prevent the model from spotting the caption from the image instead of learning visual semantics. While text-spotting filtering removes images that contain text, their CLIP-score values tend to be high. To resolve this problem, Maini et al. (2023) introduce T-MARS, a data filtering technique which aims to compute more accurate CLIP-score values, by simply masking the text (if it exists) from all images before computing the CLIP-score values. Wang et al. (2023) propose a multi-step algorithm which clusters the image embeddings, randomly samples from the clusters, and finally refines the captions of the retained samples. In contrast to their approach, we use cluster complexity to determine the number of examples to pick from each clusters; further, we pick the hardest examples from each cluster instead of random ones. In our experiments, we found that both choices improve performance.

## 3 METHODS

Our filtering pipeline has 3 stages: deduplication, CLIP-score filtering, and Density-Based-Pruning.

**Deduplication.**    We find that clusters in web-scale datasets are dominated by duplicates, not allowing us to meaningfully interpret the distance to a cluster centroid as sample difficulty. Therefore, we first deduplicate the dataset using the SemDeDup method proposed by Abbas et al. (2023), see Appendix A for details.

**CLIP-score filtering**    In CLIP-score filtering, one calculates image and caption embeddings using a strong pretrained CLIP model and removes examples below a certain cosine similarity (0.3 in LAION-400M, Schuhmann et al., 2021a) or picks a portion of the dataset with the highest cosine similarity. CLIP-score filtering removes low quality samples where the images and the captions do not match and is an integral part of many state-of-the-art pruning methods, such as Maini et al. (2023); Radenovic et al. (2023).

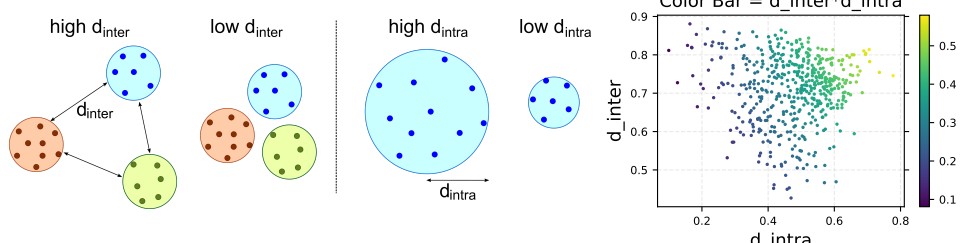

Figure 2: We determine the complexity of concepts within a dataset by examining the clusters in the embedding space of a pretrained model. We characterize the clusters with their inter-cluster (left) and intra-cluster distance (middle). We find that clusters with small inter-cluster distance tend to show similar concepts and have low variability among each other. Further, we observe that dense clusters show higher similarity among their samples. Thus, to obtain a more diverse dataset with high variability and low redundancy, we need to sample more from clusters with high inter-cluster distance and high intra-cluster distance. The scatter plot (right) shows the distribution of $d_{intra}$ over $d_{inter}$ on LAION-50M for 500 clusters.

**Density-Based Pruning (DBP)** Sorscher et al., 2022 proposed a self-supervised pruning metric for ImageNet (SSP-Pruning) where more prototypical examples are removed. We build our Density-Based Pruning (DBP) method on top of SSP-Pruning. Following Sorscher et al., 2022, we embed the data using a pretrained vision model and then cluster the data in the embedding space using `k-means` clustering into $k$ clusters. Then, considering the cluster centroid as a prototype, the method ranks the cluster items by similarity to the centroid (prototype) and removes examples with high similarities (prototypical examples).

The authors of SSP-Pruning observed that naive pruning of easy examples across the whole dataset results in strongly increasing class imbalance and degraded performance. As a solution, they introduced a class balancing score which enforced a minimum number of images per class. In the absence of class labels, a fixed cluster balancing score was used. Instead of a fixed score, we here propose to gauge the complexity of a cluster based on simple metrics to decide how many samples to keep from each cluster. To determine the complexity of clusters, we calculate the average intra-cluster distance $d_{intra}$ as the average distance of cluster members to the centroid (Fig. 2, middle) and the inter-cluster distance $d_{inter}$ as the distance of a cluster centroid to its neighboring clusters (Fig. 2, left). Intuitively, to cover the dataset optimally and to equalize the sample density across the embedding space, we need fewer samples from dense clusters and from clusters which have other clusters close nearby. Thus, we define the complexity for the j-th cluster $C_j$ as

$$C_j = d_{inter,j} \cdot d_{intra,j}, \tag{1}$$

where $d_{inter}$ is computed for each cluster $j$ as the average cosine distance between a cluster centroid and its $l$ nearest neighbor centroids, and $d_{intra}$ is computed as the average cosine distance between the items of a cluster and its centroid. We set the value of $l$ for computing $d_{inter}$ to 20 in all experiments (see Section 5.4 for an ablation over $l$). Clusters with high $d_{inter}$ and high $d_{intra}$ are considered more complex than clusters with either one of the distances being low. To enable sampling, we turn Eq.1 into a probability distribution by applying a softmax function:

$$P_j = \frac{\exp(C_j/\tau)}{\sum_i^k \exp(C_i/\tau)}, \tag{2}$$

with the temperature $\tau$ and the number of clusters $k$. We set the value of $\tau$ to 0.1 in all experiments (see Section 5.4 for an ablation over $\tau$). Multiplying $P_j$ with the target dataset size $N$, we obtain the number of examples we would like to keep from each cluster. However, it can happen that the original number of samples $M_j$ in a cluster is smaller than the desired $P_j \cdot N$. We wish to sample as close as possible to $P_j$ while honoring the dataset constraints and solve this optimization problem using a simple quadratic program solver `qpsolvers` (Caron et al., 2023). We include more details on k-means clustering and the quadratic optimization problem in Appendix B and C, respectively, and python code for solving the quadratic program and calculating $d_{inter}$ and $d_{intra}$ in Appendix C.1. The pruned cluster sizes vs $P_j N$ are plotted in Fig. 7 in the Appendix.

## 4 EXPERIMENT DESIGN

**Training Datasets.** We report results on three different datasets:

1. LAION-CAT-440M: (Radenovic et al., 2023) proposed a caption complexity, action, and text spotting filtering (CAT) method and filter the LAION-2B dataset to 440M examples (LAION-CAT-440M). We use SemDeDup (Abbas et al., 2023) to reduce the size of this dataset to 280 million examples, and call it LAION-DeDup-280M. We refer the reader to (Radenovic et al., 2023) for more details about the LAION-CAT-440M dataset. For safety purposes, we blur all human faces in the LAION-CAT-440M dataset.

2. LAION-50M: a random subset from LAION-DeDup-280M. We use this dataset mainly for development and hyperparameter search.

3. DataComp Medium dataset (Gadre et al., 2023): Since the LAION-CAT-440M dataset has already been pre-filtered in multiple ways, we complement our results on LAION by using a raw dataset with no filtering applied to it. We choose to use the DataComp Medium dataset which consists of 128 million raw examples. Because of link failures we were able to download 120 million examples from DataComp.

**Pruning the LAION dataset.** For all experiments on LAION, we focus on the training cost we save. Thus, we follow a fixed and simple setting of filtering the dataset to 60% of its original size after deduplication. Therefore, we prune LAION-DeDup-280M and LAION-50M to 166M and 30M examples, respectively. For LAION-DeDup-280M, we also experiment with pruning to 28% and 40% of its original size. Unless stated otherwise, we train for 32 epochs. For our Density-Based Pruning method, we use image embeddings from a distilled DINOV2-L/14 model (Oquab et al., 2023). We find that using the distilled DINOV2-L/14 embeddings works better than using multimodal embeddings as discussed in Section 5. We tune the number of clusters for k-means on LAION-DeDup-280M and use $k$=500 (see Section 5.4).

**Pruning the DataComp Medium dataset.** For all experiments on DataComp, we follow the protocol set by the benchmark and train for 128 million examples seen. Keeping the number of examples seen fixed means that if the dataset size decreases, the number of epochs increases. Thus, the goal here is not to reduce the training cost but to maximize performance with a fixed cost. Similar to LAION, we embed the images using the distilled DINOV2-L/14 image encoder. We tune the number of clusters on DataComp and use the value of $k$=100 as the best value.

**Pretrained encoders** DBP requires clustering and ranking examples in an embedding space of a pretrained model. We experiment with different choices and present an overview of the tested encoders in Appendix D.

**Evaluation** We use zero-shot accuracy for all evaluations and report the top-1 zero-shot accuracy on ImageNet in addition to the DataComp evaluation protocol and evaluate on a suite of 38 image classification and retrieval tasks including the VTAB tasks (Zhai et al., 2019b), ImageNet distribution shift tasks, and retrieval tasks. All the evaluation datasets we use are listed in Table 10.

**CLIP-score Baselines** We use the standard CLIP-score filtering protocol for each dataset. We use the LAION CLIP-score values from the metadata (computed using OpenAI's CLIP-B/32 model) and OpenAI's CLIP-L/14 score for DataComp.

**Other Hyperparameters** We train the CLIP-ViT-B/32 models using the OpenCLIP (Ilharco et al., 2021) default hyperparameters for both LAION and DataComp datsets and fix the training seed. We list the values of different hyperparameters in Table 6, Appendix E.

## 5 RESULTS

Our Results section is organized as follows. We first report our best results, obtained on LAION-CAT-440M (Section 5.1) and DataComp Medium (Section 5.2). In Sections 5.3 and 5.4, we analyze our results, explain our hyperparameter and design choices, and conduct ablation studies.

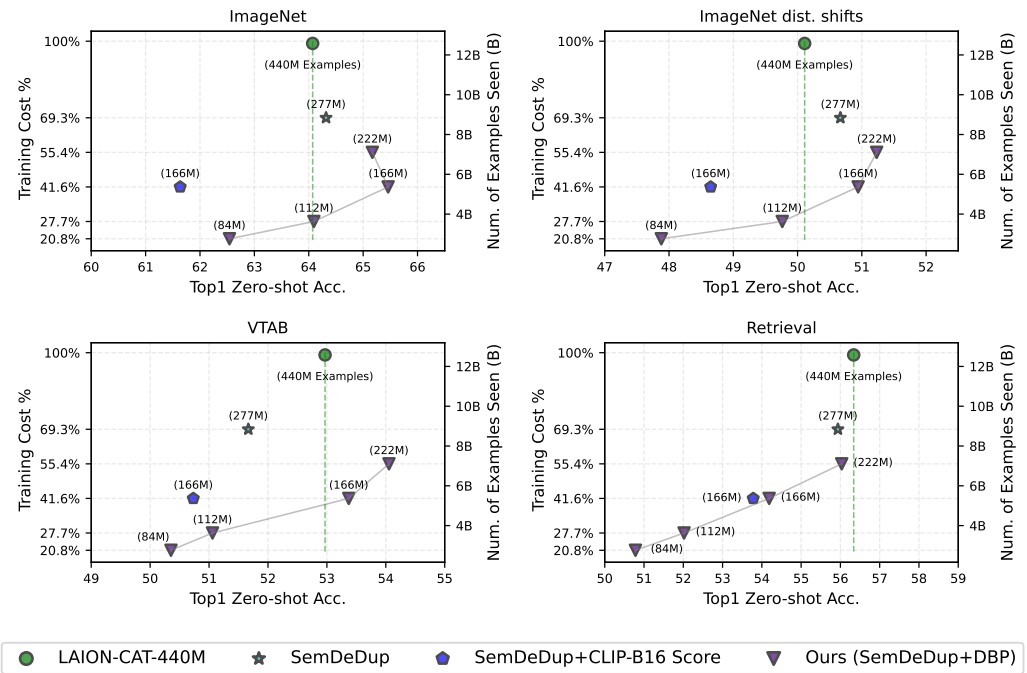

Figure 3: CLIP-ViT-B/32 zero-shot evaluation for filtering the LAION-CAT-440M dataset (Radenovic et al., 2023). We filter the data by first deduplicating it to 277M examples to get LAION-DeDup-280M (SemDeDup in the Fig.). Then we apply the DBP method to filter the LAION-DeDup-280M dataset. We see that we outperform training on the whole LAION-CAT-440M dataset on ImageNet, VTAB, and ImageNet distribution shifts datasets while using only 27%-41% of the training cost. For the LAION-CAT-440M baseline (green line), we train for 12.7B examples seen during training following the OpenAI CLIP training procedure (Radford et al., 2021a). For all other models, we train for 32 epochs regardless of the dataset size. The y-axis shows the training cost and the number of examples seen for each individual model. See Table 9 for performance details on individual datasets.

## 5.1 LAION: OUR METHOD OUTPERFORMS OPENCLIP ON IMAGENET WITH 27% OF THE TRAINING COMPUTE.

We start from LAION-CAT-440M, deduplicate it to LAION-DeDup-277M, and finally, apply the DBP method to obtain four much smaller datasets of sizes 84M, 112M, 166M, and 222M. We observe that by training on our smaller curated datasets for fewer number of iterations we can achieve better performance than training on the whole LAION-CAT-440M dataset. We show the zero-shot performance on ImageNet, ImageNet distribution shit, Retrieval, and the VATB tasks in Fig. 3 and Table 9. We observe performance gains despite the massive reduction in training compute: training on the 112M subset outperforms OpenCLIP-B/32 on ImageNet (65.44% vs 62.92%) while using only 27% of the training cost. On ImageNet distribution shit tasks and VTAB tasks, we outperform the OpenCLIP baseline using less than 41% of the training cost. On retrieval tasks, we show competitive performance despite using 55.4% of the training cost. We show detailed results for zero-shot evaluation on 38 downstream tasks in Table 9, Appendix.

## 5.2 OUR METHOD OUTPERFORMS THE CURRENT STATE OF THE ART ON DATACOMP MEDIUM WITH A LOWER DATASET SIZE.

Our approach outperforms the recently proposed and current state of the art on the DataComp leaderboard (T-MARS; Maini et al., 2023) on three (ImageNet, VTAB, and Retrieval) out of four downstream tasks families, as shown in Table 1, while T-MARS performs better on the ImageNet distribution shifts tasks. Detailed results on all shifts are shown in Table 10, Appendix. We compare our detailed results to the best baseline released by DataComp (Image-based ∩ CLIP Score (L/14 top 30%)) and report improved performance in 35 out of 38 distribution shifts. Unfortunately, the

Table 1: Our approach outperforms the current state of the art on DataComp Medium (T-MARS) on most tasks.

| Method | Size | ImageNet | ImageNet dist. shifts | VTAB | Retrieval | Average |
|---|---|---|---|---|---|---|
| TMARS (Maini et al., 2023) | 25M | 33.00 | 27.00 | 36.30 | 22.50 | 36.10 |
| Image-based ∩ CLIP Score (L/14 top 30%) (Gadre et al., 2023) | 14M | 29.70 | 23.90 | 34.60 | 23.10 | 32.89 |
| CLIP Score (L/14 top 30%) | 38M | 27.30 | 23.00 | 33.80 | 25.10 | 32.80 |
| Ours (DeDup, 80% + CLIP-L/14 Score, 50% + DBP) | 19.2M | 33.35 | 24.73 | 37.26 | 26.82 | 34.52 |
| Ours (DeDup, 80% + CLIP-L/14 Score, 40% + DBP) | 19M | 32.02 | 25.74 | 37.26 | 26.80 | 35.35 |

Table 2: DBP outperforms CLIP score filtering across different model sizes on LAION-50M. All models are trained for 5 epochs.

| Method/Model | Dataset Size | CLIP-S/32 (63M params.) IN top1 acc | CLIP-B/32 (151M params.) IN top1 acc | CLIP-L/14 (428M params.) IN top1 acc |
|---|---|---|---|---|
| CLIP score | 30M | 32.32 | 38.07 | 47.61 |
| DB-Pruning | 30M | 39.04 | 43.41 | 53.21 |

authors of T-MARS have not released their models or the performance on the individual test sets, so we cannot compare our detailed results to theirs. To achieve this result, we deduplicate the 128M examples of the DataComp dataset and retain 80% (96M) of the original dataset size, then we perform CLIP-L/14 score filtering to further reduce the dataset size to 40% (38M) or 50% (48M) of the deduplicated dataset size, and finally, we perform Density-Based Pruning (DBP) and reduce the dataset size down to around 19M examples, see Fig. 8 (Appendix) for an ablation on the optimal final dataset size as well as on the influence of the number of clusters.

## 5.3 ANALYSIS

**A smaller, more balanced dataset can lead to better models (Fig. 3).** In this work, we reduce the dataset size while maintaining and/or improving the quality of the data by balancing the data clusters and removing easy examples. This increases the marginal information gain the model gets from every training batch. As a result, we observe better performance on a variety of distribution shift tasks with shorter training: The model trained on the SemDeDup+DBP-222M dataset *almost* matches or outperforms training on the full LAION-CAT-440M dataset in all categories in Fig. 3, despite using only half the compute. This result suggests that, given a source dataset, we can find a smaller, high-quality dataset through careful filtering. Such a dataset not only enhances or maintains performance but also reduces the training cost significantly. Another works like Arjovsky et al. (2023) also shows theoretically and practically that balancing the data by "removing" examples from the majority groups/classes can result in a better worst group/class performance and a better model even though the dataset size is reduced.

**The performance on retrieval and ImageNet distribution shifts is relatively lower compared to ImageNet zero-shot accuracy (Fig. 3 and Table 1).** This trend is consistent across different baselines for retrieval tasks and we hypothesize that retrieval and ImageNet dist. shift tasks need relatively longer training (in Fig. 3 we reduce the number of training iterations/cost seen to ≤ 55.4%). To study this behavior, we measure the performance gains obtained with longer training. We increase the number of iterations for training on the 166M dataset from 41.6% (Fig. 3) to 69% and measure the difference in performance on each of the validation tasks. We observe that among the four tasks (ImageNet, ImageNet dist. shift, VTAB, retrieval), the ImageNet dist. shift and retrieval tasks benefit the most from longer training: They each gain 0.9p.p. and 0.8p.p., respectively. In contrast, ImageNet and VTAB gain 0.4p.p. and 0.7p.p., respectively. Therefore, we conclude that the observed performance drops on ImageNet dist. shifts and retrieval tasks can be, partially, attributed to shorter training.

**Our results hold across different model sizes, Table 2** We test our best approach on LAION-50M using models with different parameter counts: We train CLIP-S/32, CLIP-B/32 and CLIP-L/14 models for five epochs on 30M examples filtered from the LAION-50M dataset using our DBP method. We find that our approach outperforms CLIP score filtering for all models we tested.

Table 3: Density-based pruning (DBP) helps improve the performance of SSP-Pruning. We dedupli­cate the LAION-CAT-440M dataset to 277M examples and then apply SSP pruning or DBP to filter the dataset to 112M or 166M examples and train CLIP-B/32 on them for 32 epochs. We report the average zero-shot performance on 38 datasets from Gadre et al. (2023). We set the cluster balancing value of SSP-Pruning method to 1.0.

| Method/ Dataset size | 112M (3.6B examples seen) | 166M (5.3B examples seen) |
|---|---|---|
| DBP | 49.8 | 51.6 |
| SSP-Pruning | 48.6 | 50.7 |

Figure 4: (**left**) Performance grows consistently with continued training and we close the gap to training on the full LAION-50M dataset when training for 45 epochs, despite only using 30M samples. We also outperform the LAION CLIP-B/16 score (CS) filtering. (**right**) Density-based pruning (DBP) helps improve the performance over SSP-Pruning (Sorscher et al., 2022). We prune the LAION-50M dataset to 30M examples and train CLIP-B/32 on it for five epochs.

**DBP outperforms SSP-Pruning (LAION).** The difference between DBP and SSP-Pruning is the choice of how many examples are taken from each cluster. In SSP-Pruning, a fixed cluster balancing score is defined while in DBP, we assess the complexity of the different clusters. We show that DBP outperforms SSP-Pruning on the LAION-CAT-440M dataset (Table 3). We also show in Fig. 4(right) the benefits of DBP over SSP-Pruning on the LAION-50M dataset for different cluster balancing ratios, and find that (a) DBP outperforms SSP-Pruning across all cluster balancing ratios, and (b) cluster balancing is not necessary for DBP since we obtain the best result at a ratio of zero.

**Modality of the embeddings and the choice of the encoders are important hyperparame­ters.** When performing k-means clustering in the embedding space, we can decide whether we use the image- and/ or caption embeddings. We explore the influence of the encoder in Fig. 5(right). We experiment with image embed­dings extracted from two different pretrained en­coders, CLIP ViT-B/16, and a distilled DINOV2-L/14 model (Oquab et al., 2023). We also ex­plore using caption embeddings from two differ­ent pretrained encoders CLIP-B/16 text encoder and the Sentence BERT model "all-MiniLM-L6-v2" introduced by (Devlin et al., 2019). In

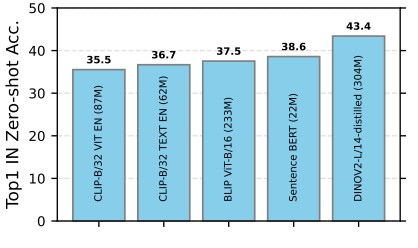

Figure 5: The choice of the encoder as well as the data modality are important hyperparameters.

addition, we use multimodal embeddings from the BLIP ViT-B/16 Image-Text Matching (ITM) head (Li et al., 2022a) which offers an elegant way to combine both modalities with a learned shared embedding. Because tuning the parameters for each model is expensive, we fixed the hyperparameters to the ones we tuned on LAION-50M using the DINOV2-L/14 embeddings. The results are displayed in Fig. 5(right) and we achieve the best results with the distilled DINOV2-L/14.

**We obtain consistent improvements with longer training, Fig. 4(left).** To show how the per­formance changes with longer training, we train the same models for five and forty-five epochs on

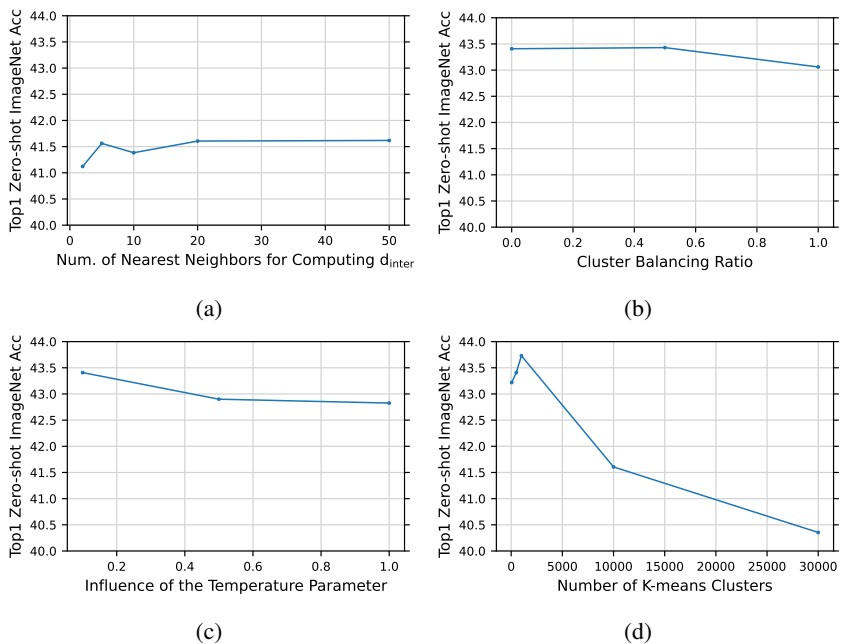

Figure 6: Values of different hyperparameters for DBP pruning. **(a)**: The number of nearest neighbors $N$ to calculate $d_{inter}$, **(b)**: the cluster balancing ratio, **(c)**: the temperature, and **(d)**: the number of clusters for k-means. We fixed the number of clusters in all experiments at 500 except for Fig. (d). We set the temperature parameters to 0.1 in all experiments except for Fig. (c). We conduct all experiments by pruning LAION-50M dataset to 30M and training on it for 5 epochs.

the LAION-50M subset, and on 30M examples filtered from it using our pipeline. We consistently outperform CLIP score filtering (CS) throughout training and even close the gap to training on the full LAION-50M dataset when training for forty-five epochs.

## 5.4 HYPERPARAMETER ABLATIONS FOR DBP

DBP has a number of hyperparameters such as the number of nearest neighbors to calculate $d_{inter}$, the cluster balancing ratio, the temperature $\tau$ in the softmax in Eq. 2, and the number of clusters for k-means clusters. The cluster balancing ratio is implemented as another constraint for the quadratic program. We choose all of these hyperparameters on LAION-50M by pruning it to 30M examples and show the results of tuning each of them in Fig. 6. Based on these results, we set the number of nearest neighbors to compute $d_{inter}$ to 20, the cluster balancing ratio to 0, the temperature to 0.1, and the number of clusters for k-means to 500.

## 6 CONCLUSION

This research accentuates the potential of refining dataset curation techniques to enhance the efficiency of model training. By challenging traditional pruning methods and incorporating the influence of proximate samples into the pruning strategy, we achieved remarkable performance improvements. Notably, on the LAION dataset, the approach surpassed the OpenCLIP-ViT-B/32 model's ImageNet zero-shot accuracy by 1.1 percentage points using merely 27.7% of the training compute. Furthermore, we report a new state of the art on the DataComp Medium benchmark for ImageNet zero-shot accuracy and impressive results across 38 evaluation tasks. This showcases the profound impact of optimized dataset pruning on the advancement of machine learning models.

## ACKNOWLEDGEMENTS

The authors would like to thank Surya Ganguli, Julian Bitterwolf, Anas Mahmoud and Roland S. Zimmermann for helpful discussions. This work was supported by the German Federal Ministry of Education and Research (BMBF): Tübingen AI Center, FKZ: 01IS18039A. The authors thank the International Max Planck Research School for Intelligent Systems (IMPRS-IS) for supporting Evgenia Rusak.

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

## A    DEDUPLICATION

We follow SemDeDup (Abbas et al., 2023) in order to deduplicate the dataset. SemDeDup deduplicates LAION by clustering the image embeddings of a pretrained model, and subsequently removing samples within a certain similarity threshold. We choose the threshold value for SemDeDup manually so that we reach the targeted dataset size. We follow the paper and keep 60%-80% of the data (63% for LAION and 80% for DataComp) as this range of values was shown to perform the best on the LAION dataset. For the k-means clustering step of SemDeDup we use 50,000 clusters for the LAION-CAT-440M dataset and 30,000 clusters for the DataComp Medium dataset. We did not tune the number cluster parameters as Abbas et al. (2023) show that it has a small effect on SemDeDup. We refer the reader to Abbas et al. (2023) for more details about the SemDeDup method.

## B    DETAILS ON K-MEANS CLUSTERING

We use the Faiss library (Johnson et al., 2019) for clustering the embeddings on a single GPU. We normalize the embeddings to have a unit length and run spherical k-means using Faiss. In all experiments, we run 100 clustering iterations. We found that 100 iterations are enough as the centroids do not change after this number of iterations.

## C    DETAILS ON THE QUADRATIC PROGRAM FOR DBP

In the main paper, we introduced a complexity criterion how to assess the complexity of individual clusters based on the distances $d_{inter}$ and $d_{intra}$. We turned the complexity into a probability distribution with a softmax function. Sampling according to this probability distribution requires solving an optimization problem, since the actual cluster sizes impose an upper bound on how many samples we can pick from each cluster. Accounting for this bound while minimizing the squared difference from the desired pruned cluster sizes, we obtain a constrained convex quadratic program:

$$\underset{x_1, x_2, \ldots, x_k}{\text{minimize}} \quad \sum_j \left( x_j^2 - 2 \cdot P_j \cdot N \cdot x_j \right)$$
$$\text{subject to} \quad \sum_j x_j = N, \ \ 1 \le x_j \le M_j \text{ for all } j, \tag{3}$$

where $x_j$ is the sampled number of examples in cluster $j$ and the constraints are given by the pruned dataset size $N$ and the actual cluster sizes $M_j$. We solve the program in Eq.3 with the publicly available quadratic program solver `qpsolvers` (Caron et al., 2023). The pruned cluster sizes vs $P_j N$ are plotted in Fig. 7.

We restate that the difference to SSP-Pruning is the replacement of the class balancing score with a method to assess the clusters' complexity to decide how many examples to keep from each cluster. Following SSP-Pruning, we also keep the least prototypical examples from each cluster.

### C.1    PYTHON CODE FOR DENSITY-BASED PRUNING

We include Python-code to solve the quadratic program defined in Eq. 3 in Table 4. The code to calculate $d_{inter}$ and $d_{intra}$ can be found in Table 5.

## D    PRETRAINED MODELS FOR CALCULATING EMBEDDINGS FOR K-MEANS CLUSTERING

**Distilled DINOV2-L/14:** We use a distilled DINOV2-L/14 model from Oquab et al. (2023). The model is distilled from DINOV2 and has 300M parameters. We resize the images of the LAION or the DataComp datasets to the size of 224x224 and take the output of the last layer of the model. Each image is embedded into a vector of size 1024.

**BLIP ViT-B/16:** We use the BLIP model to generate a multimodal representation of each image-caption pair in the data. We use the BLIP ViT-B/16 model introduced in Li et al. (2022a). The

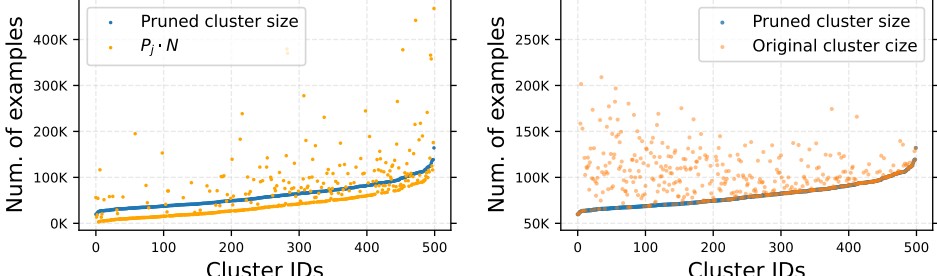

Figure 7: (left) Pruned cluster size vs $P_j N$ after solving the quadratic program. (right) Pruned cluster size vs original cluster size. We observe that the method tends to remove more examples from large clusters resulting in a more cluster-balanced dataset. In both plots, the clusters are sorted in the x-axis by the pruned cluster size. The plots are for filtering the LAION-50M dataset down to 30M examples using the distilled DINOV2-L/14 embeddings.

model has 233M parameters and has been pretrained on a dataset of 129M examples. To embed an image-caption pair, we first embed the image using the Image Encoder of BLIP into a vector of size 768. Then we condition the Image-Grounded Text Encoder of the model on the image embedding and embed the caption. We take the average of the token embeddings (each of size 768) of the last layer of the model as an embedding.

**Sentence BERT:** Sentence-BERT is a siamese BERT architecture introduced in Devlin et al. (2019). Our motivation behind using this model is the fact that the model learns to maximize the cosine similarity between embeddings of semantically meaningful sentences using a contrastive learning objective. Namely, we use the "all-MiniLM-L6-v2" Sentence BERT model from HuggingFace. This model has been trained on 1B sentence pairs dataset. The model maps each caption onto a 384-dimensional vector. This vector is the output of an average pooling layer applied on top of the last layer of the BERT model.

**CLIP ViT-B/16 Encoder** We embed the images using OpenAI's CLIP-B/16 model (Radford et al., 2021a) by mapping each image into a 512-dimensional vector using the Vision Transformer Encoder of the CLIP model. This vector is the representation of the CLS token in the output layer of the model.

**CLIP B/16 Text Encoder** We embed the captions using OpenAI's CLIP-B/16 (Radford et al., 2021a) model by mapping each caption into a 512-dimensional vector using the Text Encoder of the CLIP model. This vector is the representation of the last token in the output layer of the model.

## E TRAINING HYPERPARAMETERS

We include the training hyperparameters in Table 6.

## F ADDITIONAL ANALYSIS FOR FILTERING THE DATACOMP DATASET

**Deduplication is a necessary precursor to DBP, Table 7** Without deduplication, the clusters found by k-means during the first step of DBP are strongly influenced by the duplicates. Then, the crucial assumption of DBP—that the distance to the cluster centroid is a meaningful quantity to measure the difficulty of a particular sample—does not hold. It is therefore unsurprising that DBP works worse without prior deduplication. We note that this deduplication step has not been necessary on ImageNet where the original SSP-Pruning results have been presented, because ImageNet is a highly curated dataset.

**CLIP-score filtering leads to better results with prior deduplication, Table 8** Applying CLIP score filtering to reduce the dataset size of DataComp Medium dataset from 120M down to 38M leads to better performance if the dataset is first deduplicated.

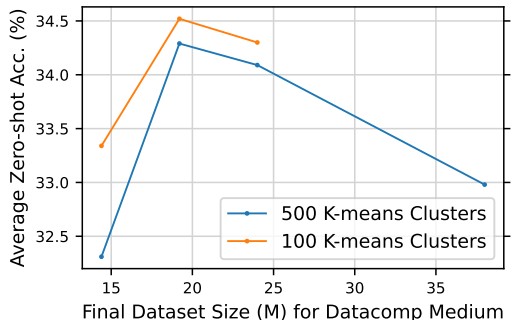

Figure 8: The performance on DataComp Medium is influenced by the dataset size as well as by the number of k-means clusters. Starting from a pool size of 120M, we first deduplicate it (to 96M), apply CLIP score filtering (to 48M), and finally apply DBP.

## G    DETAILED RESULTS ON DATACOMP MEDIUM

In addition to the averaged results in Table 1 in the main paper.   We compare our results to the best baseline model released by DataComp and take the results from the csv files at `github.com/mlfoundations/open_clip/blob/main/docs/openclip_retrieval_results.csv` and `github.com/mlfoundations/open_clip/blob/main/docs/openclip_classification_results.csv`. We show detailed results for models trained on LAION-CAT-440M and on filtered versions of this dataset in Table 9.

**Zero-shot Evaluation**   We strictly follow the evaluation protocol set up by DataComp on 38 evaluation tasks, including *ImageNet*, ImageNet distribution shit tasks (*ImageNet Sketch, ImageNet v2, ImageNet-A, ImageNet-O, ImageNet-R, and ObjectNet*), retrieval tasks (*Flickr* and *MSCOCO*), the VTAB tasks (*Caltech-101 , CIFAR-100, CLEVR Counts, CLEVR Distance, Describable Textures, EuroSAT, KITTI Vehicle Distance, Oxford Flowers-102, Oxford-IIIT Pet, PatchCamelyon, RESISC45, SVHN, and SUN397*), and other tasks.  All evaluation datasets are shown in Table 10. Detailed information on the evaluation tasks can be found in Section N of the DataComp paper (Gadre et al., 2023).

## H    SOFTWARE STACK

We use different open-source software packages for our experiments, most notably SLURM (Yoo et al., 2003), OpenCLIP (Ilharco et al., 2021), scipy and numpy (Virtanen et al., 2020), GNU parallel (Tange, 2011), Faiss (Johnson et al., 2019), PyTorch (Paszke et al., 2017) and torchvision (Marcel & Rodriguez, 2010).

Table 4: Python code for the quadratic program solver

```
1
2 import numpy as np
3 import torch
4 from qpsolvers import solve_qp
5
6 # Input: d_inter (List), d_intra (List), temp (float), num_centroids (int
      ), filtered_dataset_size (int), num_items_in_each_cluster (List)
7
8 # Output: X (list) <- Number of samples per cluster
9
10 softmax = torch.nn.Softmax()
11 probs = softmax( (d_inter *  d_intra)/temp )
12 P = np.eye(num_centroids)
13 q = - probs * filtered_dataset_size
14 A = np.array(1.0 * num_centroids)
15 b = np.array([filtered_dataset_size])
16
17 # Define the lower and upper bounds
18 min_samples = 1
19 bounds = np.array([ ( min_samples, num_items_in_each_cluster[i] )
20                       for i in range(num_centroids) ]
21
22 X = solve_qp(P=P, q=q, A=A, b=b,
23              lb=bounds[:,0], ub=[:,1], solver='osqp')
24
25 X = np.rint(X).astype(int)
```

Table 5: Python code for computing $d_{inter}$ and $d_{intra}$.

```
1
2 import numpy as np
3 import faiss
4
5 # Input: norm_embs (array), emb_dim (int), num_centroids (int),
      filtered_dataset_size (int), niter (int), seed (int), num_NNs (int)
6
7 # Output: d_intra (list), d_inter (list)
8
9 # Cluster the data
10 kmeans = faiss.kmeans(dim, num_centroids, niter=niter, seed=seed,
11                       spherical=True, gpu=True, verbose=True)
12 kmeans.train(norm_embs)
13
14 # Compute d_intra
15 sim_to_centroid, nearest_cent = kmeans.index.search(norm_embs, 1)
16
17 d_intra = []
18 for cluster_id in range(num_centroids):
19     cluster_item_ids = np.where( nearest_cent==cluster_id )
20     cluster_d_intra = ( 1 - sim_to_centroid[cluster_item_ids] ).mean()
21     d_intra.append(cluster_d_intra)
22
23 # Compute d_inter
24 sim_to_NN_centroids = kmeans.index.search( kmeans.centroids, num_NNs+1 )
25 dist = 1 - sim_to_NN_centroids[:, 1:]
26 d_inter = np.mean( dist, axis=1 )
```

Table 6: Training parameters for CLIP. We follow the standard hyperparameters used for each dataset. We use the OpenCLIP hyperparameters for experiments on the LAION dataset and the DataComp hyperparameters for experiments on the DataComp Medium dataset.

| Parameter | Value |
|---|---|
| Model | CLIP ViT-B-32 |
| Warmup (LAION) | 2000 training steps |
| Warmup (DataComp) | 500 training steps |
| Batch size (LAION) | 33,792 |
| Batch size (DataComp) | 4,096 |
| Learning rate | 5.0e-4, cosine scheduler |
| Optimizer | AdamW, wd=0.2, betas=(0.9, 0.98), eps=1.0e-6 |

Table 7: DBP is more effective on deduplicated vs non-deduplicated DataComp Medium. The result holds for two different dataset sizes. We did not apply CLIP score filtering in this experiment.

| Deduplicated? | DBP, Dataset Size (M) | ImageNet top-1 acc. | Average acc. |
|---|---|---|---|
| No | 48M | 20.70 | 27.00 |
| Yes, to 80% | 48M | 21.34 | 27.71 |
| No | 24M | 19.10 | 26.10 |
| Yes, to 80% | 24M | 20.16 | 27.30 |

Table 8: CLIP score filtering is more effective after deduplication on DataComp Medium.

| DeDup. | Pool Size | Dataset Size | ImageNet top-1 acc. | average acc. |
|---|---|---|---|---|
| No | 128M | 38M | 27.30 | 32.80 |
| Yes, to 80% | 120M | 38M | 27.93 | 33.03 |

Table 9: Evaluation Results on 38 datasets for training CLIP-B/32 models on different datasets filtered from the LAION-CAT-440M dataset. Datasets are grouped following Gadre et al. (2023). Models are evaluated using the DataComp Gadre et al. (2023) evaluation pipeline, and the *main metric* values defined by DataComp are reported in the table.

| | | Metric | OpenCLIP | LAION-440M | SemDeDup | DBP | SemD.+CS | DBP | SSP | DBP | SSP |
|---|---|---|---|---|---|---|---|---|---|---|---|
| | Datset Size | | 400M | 440M | 277M | 222M | 222M | 166M | 166M | 112M | 112M |
| | Num. Samples Seen | | 12.8B | 12.7B | 8.8B | 7.1B | 7.1B | 5.3B | 5.3B | 3.6B | 3.6B |
| | Training Cost % | | 100% | 99.2% | 69.3% | 55.4% | 55.4% | 41.6% | 41.6% | 27.7% | 27.7% |
| IN | ImageNet 1k | Acc | 62.93 | 64.07 | 64.32 | 65.17 | 61.64 | 65.46 | 65.09 | 64.09 | 62.77 |
| IN Dist. Shift | ImageNet Sketch | Acc | 49.38 | 49.78 | 49.88 | 49.5 | 47.2 | 49.21 | 49.18 | 47.36 | 46.76 |
| | ImageNet v2 | Acc | 55.06 | 55.89 | 56.11 | 56.77 | 53.14 | 57.62 | 56.79 | 56.0 | 54.94 |
| | ImageNet-A | Acc | 21.72 | 25.04 | 25.65 | 27.23 | 23.48 | 26.92 | 26.25 | 25.83 | 22.71 |
| | ImageNet-O | Acc | 53.45 | 50.6 | 51.85 | 52.5 | 52.7 | 53.05 | 55.25 | 54.6 | 55.0 |
| | ImageNet-R | Acc | 73.42 | 72.25 | 73.0 | 72.09 | 69.9 | 71.33 | 71.81 | 68.77 | 67.27 |
| | ObjectNet | Acc | 43.87 | 47.1 | 47.5 | 49.3 | 45.46 | 47.53 | 47.6 | 46.02 | 43.67 |
| VTAB | Caltech-101 | Acc | 91.18 | 90.38 | 89.97 | 90.28 | 89.01 | 90.21 | 90.81 | 89.91 | 89.74 |
| | CIFAR-100 | Acc | 70.29 | 76.48 | 75.47 | 75.33 | 73.88 | 75.97 | 76.03 | 75.31 | 73.37 |
| | CLEVR Counts | Acc | 16.24 | 23.57 | 17.21 | 33.23 | 22.7 | 25.44 | 18.37 | 21.49 | 19.97 |
| | CLEVR Distance | Acc | 23.91 | 14.97 | 17.88 | 24.51 | 22.59 | 20.45 | 18.12 | 23.77 | 24.48 |
| | Describable Textures | Acc | 54.57 | 54.2 | 54.79 | 56.06 | 48.46 | 53.94 | 52.61 | 46.17 | 46.44 |
| | EuroSAT | Acc | 51.43 | 52.72 | 44.37 | 55.76 | 47.07 | 59.56 | 47.65 | 44.69 | 41.15 |
| | KITTI Vehicle Distance | Acc | 28.97 | 13.92 | 9.56 | 17.02 | 14.06 | 25.74 | 10.97 | 23.77 | 23.49 |
| | Oxford Flowers-102 | Acc | 66.18 | 62.66 | 63.54 | 64.59 | 60.98 | 65.84 | 65.71 | 67.47 | 68.91 |
| | Oxford-IIIT Pet | Acc | 86.71 | 87.33 | 88.56 | 88.18 | 84.74 | 88.36 | 88.9 | 87.97 | 87.51 |
| | PatchCamelyon | Acc | 55.91 | 58.69 | 55.27 | 49.98 | 49.86 | 49.04 | 49.89 | 49.53 | 49.97 |
| | RESISC45 | Acc | 54.54 | 58.51 | 59.27 | 58.35 | 56.6 | 59.52 | 52.3 | 52.49 | 45.06 |
| | SVHN | Acc | 30.39 | 28.28 | 29.49 | 22.85 | 24.36 | 12.74 | 11.84 | 16.84 | 7.05 |
| | SUN397 | Acc | 66.99 | 66.87 | 66.29 | 66.55 | 65.21 | 67.03 | 66.86 | 64.36 | 63.52 |
| Retrieval | Flickr | Recall | 70.21 | 76.04 | 76.37 | 76.74 | 73.86 | 75.25 | 75.21 | 73.15 | 72.81 |
| | MSCOCO | Recall | 43.93 | 48.06 | 48.86 | 48.44 | 45.76 | 48.38 | 46.98 | 45.65 | 44.24 |
| | WinoGAViL | Jaccard Score | 40.8 | 44.91 | 42.58 | 42.92 | 41.72 | 38.93 | 37.8 | 37.26 | 36.81 |
| Others | CIFAR-10 | Acc | 90.74 | 93.75 | 93.79 | 93.82 | 92.93 | 93.28 | 93.68 | 92.31 | 92.41 |
| | Country211 | Acc | 14.75 | 14.81 | 14.78 | 15.64 | 13.8 | 14.75 | 14.0 | 13.78 | 12.68 |
| | FGVC Aircraft | Acc | 16.58 | 12.42 | 13.79 | 14.47 | 11.34 | 14.01 | 13.24 | 17.21 | 11.85 |
| | Food-101 | Acc | 80.86 | 81.29 | 81.46 | 82.41 | 79.13 | 82.55 | 80.2 | 79.98 | 75.25 |
| | GTSRB | Acc | 41.99 | 35.61 | 44.98 | 30.58 | 38.58 | 24.11 | 31.88 | 20.17 | 20.86 |
| | MNIST | Acc | 37.33 | 37.23 | 34.03 | 23.79 | 29.22 | 16.63 | 14.49 | 9.17 | 11.09 |
| | Pascal VOC 2007 | Acc | 75.82 | 79.51 | 79.77 | 80.37 | 77.96 | 79.17 | 78.88 | 78.72 | 78.59 |
| | Rendered SST2 | Acc | 52.28 | 49.53 | 49.37 | 48.16 | 52.28 | 50.25 | 48.54 | 49.97 | 47.17 |
| | Stanford Cars | Acc | 79.26 | 74.65 | 75.41 | 74.39 | 73.81 | 66.35 | 66.35 | 55.11 | 60.71 |
| | STL-10 | Acc | 95.6 | 95.73 | 96.9 | 96.3 | 96.39 | 96.28 | 96.14 | 95.71 | 95.36 |
| | iWildCam | Acc | 7.44 | 8.08 | 7.28 | 7.86 | 7.27 | 8.82 | 7.57 | 7.17 | 5.0 |
| | Camelyon17 | Acc | 47.04 | 50.2 | 62.71 | 49.84 | 50.0 | 48.91 | 50.1 | 49.6 | 49.84 |
| | FMoW | Acc | 12.96 | 12.38 | 14.75 | 13.96 | 14.12 | 0.0 | 8.87 | 0.0 | 0.0 |
| | Dollar Street | Acc | 54.91 | 58.41 | 56.43 | 57.94 | 55.84 | 55.37 | 55.96 | 57.48 | 55.84 |
| | GeoDE | Acc | 83.8 | 86.01 | 84.76 | 84.72 | 84.01 | 84.22 | 84.76 | 84.72 | 83.76 |
| | Average | | 52.72 | 52.95 | 53.11 | 53.09 | 51.34 | 51.64 | 50.7 | 49.83 | 48.63 |

Table 10: Evaluation results on 38 datasets for training CLIP-B/32 models on filtered DataComp Medium (120M examples). Datasets are grouped following Gadre et al. (2023). Note that for DataComp all models are trained for the same number of examples seen following the DataComp training settings.

| | Dataset | Metric | Image-based ∩ CLIP-score (L/14 top 30%) (Gadre et al., 2023) | DeDup,80% +CLIP Score,40% +DBP | DeDup,80% +CLIP-score,50% +DBP |
|---|---|---|---|---|---|
| | Num. samples seen | | 120M | 120M | 120M |
| IN | ImageNet 1k (Deng et al., 2009) | Acc | 29.72 | 32.02 | **33.35** |
| IN Dist. Shift | ImageNet Sketch (Wang et al., 2019) | Acc | 19.3 | **20.17** | 17.73 |
| | ImageNet v2 (Recht et al., 2019) | Acc | 24.4 | 26.54 | **28.34** |
| | ImageNet-A (Hendrycks et al., 2021b) | Acc | 4.93 | 5.57 | **6.45** |
| | ImageNet-O (Hendrycks et al., 2021b) | Acc | 40.85 | **43.75** | 42.7 |
| | ImageNet-R (Hendrycks et al., 2021a) | Acc | 34.02 | **35.46** | 32.78 |
| | ObjectNet (Barbu et al., 2019) | Acc | 19.71 | **22.95** | 20.36 |
| VTAB | Caltech-101 (Fei-Fei et al., 2004) | Acc | 71.59 | **71.74** | 70.97 |
| | CIFAR-100 (Krizhevsky et al., 2009) | Acc | 54.76 | 58.54 | **59.34** |
| | CLEVR Counts (Johnson et al., 2017; Zhai et al., 2019a) | Acc | 13.65 | 14.58 | **16.38** |
| | CLEVR Distance (Johnson et al., 2017; Zhai et al., 2019a) | Acc | 22.49 | 22.37 | **23.62** |
| | Describable Textures (Cimpoi et al., 2014) | Acc | 21.33 | 22.18 | **23.35** |
| | EuroSAT (Helber et al., 2019; Zhai et al., 2019a) | Acc | 33.93 | **35.98** | 34.22 |
| | KITTI Vehicle Distance (Geiger et al., 2012; Zhai et al., 2019a) | Acc | 21.1 | 30.8 | **36.57** |
| | Oxford Flowers-102 (Nilsback & Zisserman, 2008) | Acc | 29.65 | 32.76 | **33.8** |
| | Oxford-IIIT Pet (Parkhi et al., 2012; Zhai et al., 2019a) | Acc | 43.11 | 45.64 | **47.22** |
| | PatchCamelyon (Veeling et al., 2018; Zhai et al., 2019a) | Acc | 58.62 | **59.58** | 48.9 |
| | RESISC45 (Cheng et al., 2017; Zhai et al., 2019a) | Acc | 27.78 | 29.83 | **30.92** |
| | SVHN (Netzer et al., 2011; Zhai et al., 2019a) | Acc | 15 | **16.97** | 14.01 |
| | SUN397 (Xiao et al., 2016) | Acc | 36.37 | 43.35 | **45.08** |
| Retrieval | Flickr (Young et al., 2014) | Recall | 18.12 | **27.46** | 26.85 |
| | MSCOCO (Chen et al., 2015) | Recall | 11.0 | **16.78** | 16.45 |
| | WinoGAViL (Bitton et al., 2022) | Jaccard score | 43.37 | 36.16 | 37.17 |
| Others | CIFAR-10 (Krizhevsky et al., 2009) | Acc | 82.52 | 84.66 | **85.91** |
| | Country211 (Radford et al., 2021b; Thomee et al., 2016) | Acc | 4.53 | 5.55 | **6.0** |
| | FGVC Aircraft (Maji et al., 2013) | Acc | 3.04 | 3.38 | **3.86** |
| | Food-101 (Bossard et al., 2014) | Acc | 41.68 | 46.99 | **49.07** |
| | GTSRB (Stallkamp et al., 2011) | Acc | **13.66** | 11.77 | 10.17 |
| | MNIST (LeCun, 1998) | Acc | 11.47 | **14.77** | 10.06 |
| | Pascal VOC 2007 (Everingham et al., 2007) | Acc | 54.59 | 67.47 | **71.83** |
| | Rendered SST2 (Zhai et al., 2019a) | Acc | **53.16** | 50.58 | 50.14 |
| | Stanford Cars (Krause et al., 2013) | Acc | 28.03 | **31.14** | 28.53 |
| | STL-10 (Coates et al., 2011) | Acc | 83.65 | 84.21 | **86.21** |
| | iWildCam (Beery et al., 2020; Koh et al., 2021) | Acc | 1.42 | **2.33** | 1.81 |
| | Camelyon17 (Bandi et al., 2018; Koh et al., 2021) | Acc | 66.69 | **73.5** | 47.72 |
| | FMoW (Christie et al., 2018; Koh et al., 2021) | Acc | 0.0 | 0.0 | 0.0 |
| | Dollar Street (Rojas et al., 2022) | Acc | 44.98 | 46.03 | **47.08** |
| | GeoDE (Ramaswamy et al., 2023) | Acc | 65.59 | **68.94** | 66.76 |
| | Average | | 32.89 | **35.35** | 34.52 |

