# OpenReview forum: "Effective pruning of web-scale datasets based on complexity of concept clusters"
_ICLR.cc/2024/Conference — ICLR 2024 poster_

### Official Review · Reviewer_orZi · 2023-10-30

**Soundness:** 3 good
**Presentation:** 3 good
**Contribution:** 3 good
**Rating:** 5
**Confidence:** 4

**Summary:**

This paper focuses on improving the training and data efficiency of CLIP-style models by pushing the limits of pruning large-scale multimodal datasets. By adapting the pruning rate to the complexity of different concepts within the dataset, the authors are able to reduce training costs to a quarter of regular training. They outperform the LAION-trained OpenCLIP-ViT-B/32 model on ImageNet zero-shot accuracy by 1.1 percentage points while using only 27.7% of the data and training compute. Additionally, they achieve a new state-of-the-art ImageNet zero-shot accuracy and competitive performance on 38 evaluation tasks in the DataComp Medium benchmark. The findings demonstrate the potential of pruning methods for improving the efficiency of training multimodal models.

**Strengths:**

1. The method proposed in this paper exhibits a certain degree of novelty.
2. The paper extensively validates its findings using a large-scale experimental dataset, although experiments with even larger models were not performed.
3. The practical applicability of the approach presented in this paper suggests its value in real-world scenarios.

**Weaknesses:**

1. We cannot determine if the current data is effective for training ViT-B-16 and ViT-L-14 models.
2. The majority of the experimental evaluation metric is Imagenet zero-shot top1. Is there any bias in the algorithm towards Imagenet data?
3. Detailed results for each dataset in VTAB are not provided.

**Questions:**

In this paper， the author scales SSP-Pruning to web-scale datasets and demonstrates that the pruning criterion can also transfer to the DataComp benchmark.
Q1: In Fig. 1, the author shows they reduced the LAION-CAT440M to 166M and improved the zero-shot performance on Imagenet, can you provide more results? such as zero-shot transfer and linear probe performance on different datasets.
Q2: Does the robustness of model training change when the dataset size is reduced? Please provide robustness evaluation results.
Q3: The difference between DBP and SSP-Pruning is the choice of how many examples are taken from each cluster. It seems that innovation has some shortcomings
Q4: In Fig.3, 222M performs worse zero shot accuracy than 166M, please give a reasonable explanation.
Q5: Why weren't models of the same size, such as CLIP ViT-L-14, OPENCLIP LAION400M ViT-L-14, EVA ViT-L-14, and DINOv2 ViT-L-14, used for clustering comparisons?
Q6:

---

> ### Author Response · Authors · 2023-11-16
> **Response to Reviewer orZi (part 1)**
>
> We thank reviewer orZi for their valuable feedback and respond to the individual concerns below.
>
> ### We cannot determine if the current data is effective for training ViT-B-16 and ViT-L-14 models.
> We have evaluated this question in our paper and would like to point the reviewer to Table 2 where we show that DBP outperforms CLIP-score filtering for different model sizes.
> We also include the Table below for convenience.
> | | Dataset | CLIP-S/32  | CLIP-B/32 | CLIP-L/14 |
> |--|-|--|---|---|
> | Method/Model | Size    | (63M params.) | (151M params.) | (428M params.) |
> |  |   | IN top1 acc   | IN top1 acc    | IN top1 acc    |
> | CLIP score   | 30M     | 32.32  | 38.07   | 47.61  |
> | DB-Pruning   | 30M     | **39.04**   |**43.41** | **53.21**  |
>
>
> ### The majority of the experimental evaluation metric is Imagenet zero-shot top1. Is there any bias in the algorithm towards Imagenet data? Detailed results for each dataset in VTAB are not provided.
> We thank the reviewer for their comment and refer them to the newly added Tables 9 and 10 in the Appendix; we also included Table 10 in our general comment to all reviewers. In Table 9, we find that we outperform the best baseline released by DataComp on 35 out of 38 tasks. T-MARS have unfortunately neither released the numbers on all test sets nor the trained models which makes it impossible to compare to their results. In Table 10, we show results on the 38 tasks from DataComp for the models trained on LAION-CAT-440M, as well as on filtered versions of this dataset.
>
>
> ### Does the robustness of model training change when the dataset size is reduced? Please provide robustness evaluation results.
> Thank you for raising this important concern. We added the following paragraph to our Results section:
>
> The performance on retrieval and ImageNet distribution shifts is relatively lower compared to ImageNet zero-shot accuracy (Fig. 3). This trend is consistent across different baselines for retrieval tasks and we hypothesize that retrieval and ImageNet dist. shift tasks need relatively longer training (in Fig. 3 we reduce the number of training iterations/cost seen to ≤ 55.4%). To study this behavior, we measure the performance gains obtained with longer training. We increase the number of iterations for training on the 166M dataset from 41.6% (Fig. 3) to 69% and measure the difference in performance on each of the validation tasks as illustrated in Fig 4(right). We observe that among the four tasks (ImageNet, ImageNet dist. shift, VTAB, retrieval), the ImageNet dist. shift and retrieval tasks benefit the most from longer training. Therefore, we conclude that the observed performance drops on ImageNet dist. shifts and retrieval tasks can be attributed to shorter training.
>
> ### The difference between DBP and SSP-Pruning is the choice of how many examples are taken from each cluster. It seems that innovation has some shortcomings.
> We would like to stress that scaling SSP-Pruning to LAION was highly untrivial in itself because 1) it required an initial step of deduplication as well as 2) the correct choice of the encoder to calculate the image / caption embeddings. We also believe that our idea to use cluster statistics to gauge the complexity of different concepts is novel and has not been proposed for dataset pruning.
>
> ### In Fig.3, 222M performs worse zero shot accuracy than 166M, please give a reasonable explanation.
>
> Let’s recall that data quality is different from data quantity. In this work, we reduce the dataset size while maintaining and/or improving the quality of the data by balancing the data clusters and removing easy examples. This increases the marginal information gain the model gets from every training batch.
> We pruning a dataset we need to balance between the dataset size and the dataset quality. The 222M dataset (lower quality) and the 166M (higher quality) are good examples of this balance for the ImageNet classification task. We notice that same performance behavior when we look at the LAION-440M (lower quality) dataset and the filtered datasets (higher quality) in Fig 3.
>
> Some previous works like, [1] show theoretically and practically that balancing the data by “removing” examples from the majority groups/classes can result in a better worst group/ class performance and a better model even though the dataset size gets reduced. We added a discussion on this point to our Results section.
>
> [1] Arjovsky et al. “Throwing away data improves worst-class error in imbalanced classification”, ICML 2023

---

> > ### Author Response · Authors · 2023-11-16
> > **Response to Reviewer orZi (part 2)**
> >
> > ### Why weren't models of the same size, such as CLIP ViT-L-14, OPENCLIP LAION400M ViT-L-14, EVA ViT-L-14, and DINOv2 ViT-L-14, used for clustering comparisons?
> >
> > We agree with the reviewer that using models with the same size can be a good way to compare between models. In this work, we want to show that the quality of the clustering embeddings doesn’t depend on the model size only. For example, the Sentence BERT model is a very small text transformer (22M param.) that has been used by [2] for clustering, and we find that it performs better than much bigger models like CLIP or BLIP which have been trained using multimodal datasets.
> >
> > [2] Tirumala el.al.: “D4: Improving LLM pretraining via document de-duplication and diversification.”

---

### Official Review · Reviewer_mnGt · 2023-10-31

**Soundness:** 2 fair
**Presentation:** 3 good
**Contribution:** 1 poor
**Rating:** 3
**Confidence:** 5

**Summary:**

This work scales SSP-Pruning to web-scale datasets, investigate how the complexity of different concepts within a dataset can be used for pruning, and report further improvements over regular SSP-Pruning. They evaluate it with CLIP pre-training and report the results on classification task: it shows data efficiency in terms of pre-training.

**Strengths:**

The paper is well structured and the motivation is clear. It is interesting to study the data efficiency of CLIP pre-training to reduce the computational resources.

**Weaknesses:**

1. The biggest concern I have is about the technical novelty. The main idea is based on SSP-Pruning. Therefore, the technical contribution is very trivial.
2. This work only reports the results on image classification. How about the zero-shot results of CLIP in retrieval tasks, such as COCO/Flickr30k image-to-text and text-to image retrieval? The zero-shot retrieval performance is also very important to measure the quality of pre-training and the effectiveness of the proposed method. The paper mentions about the retrieval task and shows some results in Figure 3 but I am not sure the implementation details: which dataset do they use? It would be helpful to report the results on standard benchmark (COCO/Flickr30k).
3. The method involves CLIP-score filtering to curate a new dataset to train CLIP. I am unsure if such involvement is fair, as the filtering itself is similar to distilling some knowledge from a well-trained CLIP. Then, the new CLIP is somehow learning from the well-trained CLIP on large-scale datasets. Therefore, it may not be fair to claim the data efficiency for the proposed method.

**Questions:**

See weaknesses.

---

> ### Author Response · Authors · 2023-11-16
> **Response to Reviewer mnGt**
>
> We thank reviewer mnGt for their valuable feedback and respond to the individual concerns below.
>
> ### The main idea is based on SSP-Pruning. Therefore, the technical contribution is very trivial.
>
> We would like to stress that scaling SSP-Pruning to LAION was highly untrivial in itself because 1) it required an initial step of deduplication as well as 2) the correct choice of the encoder to calculate the image / caption embeddings. We also believe that our idea to use cluster statistics to gauge the complexity of different concepts is novel and has not been proposed for dataset pruning.
>
> ### This work only reports the results on image classification. How about the zero-shot results of CLIP in retrieval tasks, such as COCO/Flickr30k image-to-text and text-to image retrieval?
> We thank the reviewer for their comment and refer them to the newly added Tables 9 and 10 in the Appendix; we also included Table 10 in our general comment to all reviewers. In Table 9, we find that we outperform the best baseline released by DataComp on 35 out of 38 tasks. T-MARS have unfortunately neither released the numbers on all test sets nor the trained models which makes it impossible to compare to their results. In Table 10, we show results on the 38 tasks from DataComp for the models trained on LAION-CAT-440M, as well as on filtered versions of this dataset.
>
> ### The method involves CLIP-score filtering to curate a new dataset to train CLIP. I am unsure if such involvement is fair, as the filtering itself is similar to distilling some knowledge from a well-trained CLIP. [...]Therefore, it may not be fair to claim the data efficiency for the proposed method.
> We thank the reviewer for raising this important concern. CLIP-score filtering has become a standard step in almost all data selection frameworks. For example, DataComp is becoming an established benchmark for testing data pruning methods at scale, and uses CLIP-score filtering as part of most of their baselines. Thus, we are comparing methods which all use CLIP-score filtering in some form. In our Table showing DataComp results and in our Figure 1 showing results on LAION, all baselines have also used CLIP-score filtering. In that sense, our approach is more efficient with respect to other methods which also use CLIP-score filtering.

---

### Official Review · Reviewer_u5qD · 2023-11-09

**Soundness:** 3 good
**Presentation:** 3 good
**Contribution:** 3 good
**Rating:** 8
**Confidence:** 4

**Summary:**

The paper proposes a dataset pruning method broadly applicable to web scale datasets such as LAION. The authors build upon important prior work at the ImageNet scale (SSP-Pruning, Sorscher et al., 2022), which attempts to rebalance a dataset by clustering, and then sampling datapoints inversely proportional to their cluster centroids. The authors improve upon this procedure by noting that clusters have varying degrees of importance, both as a result of the average distance of points from the centroid as well as the average distance of the centroid from other centroids, and determine the proportion of samples to be pruned proportional to the importance. The authors demonstrate the supremacy of this method, DPB, through an array of experiments that the pruned datasets jointly save on compute while increasing overall performance.

**Strengths:**

The paper's extensive analysis of their new method across multiple datasets and benchmarks makes a compelling case for the high performance of their method. The dimensions of ablations, and the high level takeaways from those ablations in the results section are clear -- DBP outperforms SSP Pruning, outperforms CLIP filtering, DBP outperforms SemDeDup alone, and Deduplicaiton is crucial to the success of the method. These takeaways make the contribution quite compelling.

**Weaknesses:**

The core weaknesses concern clarity of the results, and the overall difficulty in connecting the many different - at times seemingly disparate - results that are thrown together in the narrative.

For one, the paper begins by proposing an improvement to SSP, but defers any result about the paper to a small table in the appendix, for a single model and single dataset -- it would help to do a more comprehensive evaluation to make clear the improvement being proposed. The authors attempt to briefly make a broader connection that their work is akin to density based pruning -- this is where the relative lack of optimization over the number of clusters -- which very much controls how the density is approximated -- is puzzling. Indeed, the 5 point plot of Figure 6-d leaves much to want, where we are led to believe 500 is the best, because it is better than 100 (?) and 10000. More ablations here would help clarify things. In general, more time assessing Density based pruning methods (e.g. for a datapoint, if its k nearest neighbors approximate it well, it can be discarded) would significantly improve the narrative of the paper.

On the other hand, the quadratic program, though interesting and unexpected in a paper of this type, seems like overkill -- assigning the maximum number of points to each cluster whose expected number of samples is above the maximum, and redistributing the remaining samples proportionally among the remaining clusters (and repeating) seems like it would achieve what the authors are seeking to do (if not exactly what the QP does) without the overhead of a QP solver, and all the space in the paper it consumes.

The discussion section would benefit from a more clear discussion of where the shortcomings of the method are relative to the other methods presented in the paper. For example, DBP does a tiny bit better than TMARS on Imagenet, but more significantly worse on Imagenet Dist and Average but a discussion of the latter point is entirely missing. Whether pruning in a density based manner affects model robustness seems like a subject at least worth touching on. Same for the retrieval plot in Figure 4 -- why is it that SemDeDup does worse than LAION-440B, but DBP catches up? An analysis of what is happening here would help understand the paper considerably.

Beyond that, the many different numbers in the paper -- as a result of different filtering models (CLIP-B16 or CLIP-L/14), differing number of epochs, differing initial dataset size (LAION-50M or 440M) -- make the results very difficult to connect and piece together across tables and figures. If some consistency across variables were afforded -- or at least enough ablations such that there is an extra column in each table connecting the numbers in other tables to that table, it would make the information much easier to process.

Update: The rebuttal does well to address most of my concerns.

**Questions:**

If the intuition is that it is density based pruning, why only 500 clusters? The 5 point plot of figure 6-d suggests that clusters between 100 and 10000 should be better explored since there are sharp peaks in that range.

Figure 5: Why not use concatenate CLIP's text + image embedding?

How is it that SemDeDup does worse than LAION-440M for VTAB, but DBP does better?

A notable decrease in performance on ImageNet dist shifts and Retreival tasks -- does the DBP type of pruning hurt generally hurt robustness?

Why use CLIP-B16 Score for Figure 3, but CLIP-L/14 Score for Table 1?

Update: thank you for taking the time to answer my questions, they are sufficiently addressed.

---

> ### Author Response · Authors · 2023-11-16
> **Response to Reviewer u5qD (part 1)**
>
> We thank reviewer u5qD for their valuable feedback and respond to the individual concerns below.
>
> ###  Is using the QP solver necessary?
> Using the QP solver provides a minimal overhead–it requires between 5 and 15 minutes on a single CPU depending on the dataset size. We agree that the relatively simple program takes an excessive amount of text in the paper; thus, we have moved the optimization problem and the algorithm to the Appendix.
>
> ### Please provide a broader ablation on the number of clusters.
> We thank the reviewer for this suggestion and ran an experiment for a more fine-grained analysis on the number of clusters. To save compute, we trained for one epoch on our 166M dataset (Fig. 3) for a wider range of values for the number of clusters hyperparameter (50, 100, 500, 1000, 5000, 10000, 20000, 60000, 10000).
>
> | Number of Clusters |50|100|500|1000|5000|10000|20000|60000|100000|
> |---|---|---|---|---|---|---|---|---|---|
> |top1 zero-shot accuracy |50.508| 51.442 | **52.338** | 52.214 | 51.764 | 50.71 | 50.278 | 49.374 | 48.01 |
>
> ### Why do you not concatenate CLIP's text + image embedding?
> We briefly tried this during the development stage of the project and found that it doesn’t work better than using the multi-modal embedding. We think that the BLIP embedding shares information between the image and caption embeddings which is missing in a simple concatenation.
>
> ### Why does SemDeDup do worse than LAION-440M on VTAB, but DBP does better? Analogously, why does DBP (166M) do better than SemDeDup (280M) or LAION-440M?
>
> Thank you for raising this concern. We added the following paragraph to our Results section: A smaller, more balanced dataset can lead to better models (Fig. 3). In this work, we reduce the dataset size while maintaining and/or improving the quality of the data by balancing the data clusters and removing easy examples. This increases the marginal information gain the model gets from every training batch. As a result, we observe better performance on a variety of distribution shift tasks with shorter training: The model trained on the SemDeDup+DBP-222M dataset matches or outperforms training on the full LAION-CAT-440M dataset in all categories in Fig. 3, despite using only half the compute. The work of Arjovsky et al. (2023) also shows theoretically and practically that balancing the data by “removing” examples from the majority groups/classes can result in a better worst group/class performance and a better model even though the dataset size is reduced.
>
> Arjovsky et al. “Throwing away data improves worst-class error in imbalanced classification”, ICML 2023
>
> ### What are DBP’s shortcomings compared to T-MARS?
> Unfortunately, the T-MARS authors have not released the numbers on the individual distribution shifts, making it impossible to perform a fine-grained comparative analysis of both methods.
>
> ### Why do we see performance drops on ImageNet dist. shifts and retrieval tasks?
> Performance on retrieval tasks is relatively lower than on the other tasks. As this trend is consistent across different baselines, we hypothesize that retrieval tasks need relatively longer training (In Fig. 3 we reduced the number of training iterations/cost seen to <=55.4\%).
>
> To study this behavior we conducted the following experiment: We compare the relative gains from longer training on the different tasks in Fig.3. In more detail, we compare the difference in performance gain when training for 41.6% or for 69% of the number of iterations compared to training on the full dataset. We show the difference in performance on each of the validation tasks in the table below. We observe that among the four tasks (ImageNet, ImageNet dist. shift, VTAB, retrieval), the ImageNet dist. shift and retrieval tasks benefit the most from longer training. Therefore, we conclude that the observed performance drops on ImageNet dist. shifts and retrieval tasks can be attributed to shorter training.
>
> | Dist. shift | relative performance gain |
> |-|-|
> |ImageNet dist. shifts | 0.9%|
> |Retrieval| 0.8%|
> |VTAB| 0.7%|
> |ImageNet| 0.4%|
>
> We added this discussion and the Table (as a bar plot) to the Results section of our revised paper.

---

> ### Author Response · Authors · 2023-11-16
> **Response to Reviewer u5qD (part 2)**
>
> ### Why do we use different architectures for filtering on LAION (CLIP-B/16) vs DataComp (CLIP-L/14)?
>
> The reason we use two CLIP models for computing the CLIP score (B/16 for LAION and L/14 for Datacomp) is to comply with the baselines used in the literature by using the meta-data clip score for each dataset. For example, SemDeDup uses CLIP-B/16 to deduplicate LAION; also, CLIP-B/16 was used to filter LAION-440M. On the other hand, DataComp chose to use CLIP-L/14 for filtering which forced us to also use this model to remain comparable to their reported baselines.
>
> It is also important to note that for a given dataset (LAION or Datacomp) we use the same CLIP filtering model for all the datasets we created and that keeps the comparison between models fair and consistent for each dataset.
>
>
> ### The results section is unclear due to inconsistencies in the choice of the encoder / training epochs etc.
>
> We thank the reviewer for this important feedback. We restructured the Results section and separated the analysis experiments on the LAION dataset (main text) and the analysis experiments on how to prune the DataComp dataset (now in Appendix F) into two sections.
>
> We thank the reviewer for providing this valuable feedback. Consequently, we have restructured the results section, segregating the analysis experiments conducted on the LAION dataset (included in the main text) from the analysis experiments on how to prune the DataComp dataset, which are now found in Appendix F.
>
> The new structure is as follows:
> - Fig.1, Fig. 3 and Table 3: FIltering the LAION-440M
> - Table 1: Filtering the DataComp dataset
> - All the remaining figures: Method analysis and hyperparameters tuning using a small dataset (LAION50M).
>
> We are happy to discuss and accommodate further requested changes with reviewer u5qD.

---

### Official Review · Reviewer_7m8Z · 2023-11-11

**Soundness:** 3 good
**Presentation:** 2 fair
**Contribution:** 2 fair
**Rating:** 5
**Confidence:** 4

**Summary:**

This paper seeks to prune large-scale multimodal datasets (e.g., LAION) for training CLIP-style models for training and data efficiency. Building upon SSP-pruning, Density-Based-Pruning picks the number of samples per cluster based on the overall complexity of a particular cluster, and achieves state-of-the-art results on LAION.

**Strengths:**

1.	This paper proposes Density-Based-Pruning to improve data efficiency on large-scale multimodal datasets.
2.	This paper conducts detailed hyperparameter selection experiments, which made the experimental results more convincing.

**Weaknesses:**

1.	The related work section contains excessive content, and there is a considerable amount of duplication. Besides, it is better to introduce “coreset selection[1-3]”, as it is closely related to your work.

2.	Missing some baselines for comparison, including random selection and other data pruning methods [3-4].

3.	The description in the methods section is very confusing.
[1] Moderate coreset: A universal method of data selection for real-world data-efficient deep learning, ICLR 2023
[2] Large-scale Dataset Pruning with Dynamic Uncertainty, arxiv23
[3] RETRIEVE: Coreset Selection for Efficient and Robust Semi-Supervised Learning, NeurIPS 2021
[4] Too Large; Data Reduction for Vision-Language Pre-Training, ICCV
2023

**Questions:**

1.	In Fig. 4(right), more sample points are needed for convincing conclusions.
2.	In Table 2, it would be better to include SSP-Pruning as a baseline for comparison.

3.	The tasks are relatively simple (ImageNet zero-hot). Could we compare DBP with other baselines on different tasks?
4.	In the methods section, the filtering pipeline includes deduplication, CLIP score filtering, and Density-Based Self-Supervised Prototypes pruning. However, there is no description about CLIP score filtering. The description about deduplication does not provide me with any useful information.

---

> ### Author Response · Authors · 2023-11-16
> **Response to Reviewer 7m8Z**
>
> We thank reviewer 7m8Z for their valuable feedback and respond to the individual concerns below.
>
> ### Missing baselines for comparison, including random selection and other data pruning methods [3-4].
> We thank the reviewer for their suggestions.
> * Random selection: We excluded the random pruning baseline to save compute because it has been proven to be very weak. For example, [1] shows that random pruning consistently lags behind the SemDeDup baseline across various dataset sizes and evaluation tasks. Random selection has been tested by the DataComp creators on the “small” scale where it underperformed all other baselines (Table 12, https://arxiv.org/pdf/2304.14108.pdf). [1] Abbas et.al. 2023 https://arxiv.org/abs/2303.09540
> * Comparison to [3]:
>     * **The task considered in [3] is not compatible with ours.** In [3], the authors tackle the problem of semi-supervised learning and select the coreset of the unlabeled data resulting in minimum labeled set loss when trained upon in a semi-supervised manner. In contrast, we consider the fully unsupervised setting where we train a CLIP model on a pre-chosen coreset.
>     * Further, **The dataset and model scales of [3] differ strongly from ours.** In [3], the authors perform experiments on small-scale datasets such as CIFAR10 and SVHN using WideResNets and LeNet architectures, while we perform experiments on web-scale datasets using transformer architectures. Sorscher et al. showed that methods which perform well on small-scale datasets do not scale to ImageNet, and we found scaling SSP-Prototypes up to LAION / DataComp highly untrivial. We hypothesize that scaling [3] to DataComp to enable a fair comparison will be highly involved, as well as it is not clear that the small-scale datasets are sufficiently large to train transformer models on.
> * Missing comparison to [4]: While the setting of [4] is comparable to ours since the authors of [4] also study CLIP training on large-scale datasets, **their choice of datasets is different from ours**. Our base datasets are either LAION-CAT-440M (which is a filtered version of LAION-2B) or DataComp Medium, while [4] conduct CLIP experiments on CC3M. To enable a fair comparison, we would either need to scale [4] to DataComp or implement our method on CC3M; neither of these options is trivial and cannot be done during the short rebuttal period. If the reviewer would like to see results of DBP on CC3M, we would be happy to run these experiments until the camera ready version.
>
> ### The description in the methods section is confusing.
> We rewrote and restructured the Methods section to improve clarity. In particular, we rewrote the paragraphs on deduplication and CLIP-score filtering. Following the suggestion of reviewer u5qD, we moved details on the QP program to the Appendix.
>
> ### The related work section contains excessive content, duplication and there are missing works which should be discussed.
> We thank the reviewer for their suggestions and apologise for missing these important papers. We added them to our related work section which now has a new paragraph titled “Data curation in supervised learning” where we included several other works in addition to the reviewer’s suggestions. We also removed redundancy between Introduction and Related Work.
>
> ### The tasks are relatively simple (ImageNet zero-shot). Could we compare DBP with other baselines on different tasks?
> We thank the reviewer for their comment and refer them to the newly added Tables 9 and 10 in the Appendix; we included Table 10 in our general comment to all reviewers. In Table 9, we find that we outperform the best baseline released by DataComp on 35 out of 38 tasks. T-MARS have unfortunately neither released the numbers on all test sets nor the trained models which makes it impossible to compare to their results. In Table 10, we show results on the 38 tasks from DataComp for the models trained on LAION-CAT-440M, as well as on filtered versions of this dataset.

---

> > ### Comment · Reviewer_7m8Z · 2023-11-20
> > **thanks for author rebuttal**
> >
> > thanks for the rebuttal. i have a question:
> > have the authors tried to use DBSCAN and then select a subset as a baseline for comparison?

---

> ### Author Response · Authors · 2023-11-20
> **Response regarding DBSCAN**
>
> Thank you for this interesting suggestion. There would be two ways to implement DBSCAN as a baseline: (1) as a replacement of DBP which is based on k-means clustering or (2) as an additional step before performing kmeans clustering to potentially remove outliers from the data. In the following, we will comment on both.
>
> ### Considering option (1):
> We could use DBSCAN instead of DBP which is based on k-means clustering. We could sample data points based on the density which we could estimate as the mean distance between cluster members. We see the following issues with this approach:
> - **Clusters obtained with DBSCAN do not offer a simple way to rank cluster examples.**  Our method (as well as other coreset selection methods, see our revised Related Work section) rely on some ranking metric to select examples to train the model on and most methods propose to take the most difficult examples. K-means has the advantage of spherically-shaped and prototype-based clusters which allows us to rank the examples by measuring their distance from the cluster centroid. DBSCAN does not provide us with cluster prototypes which we could use to rank examples. While we could define cluster centroids as the mean or median of all cluster members, we are doubtful how accurate or useful these estimates would be, given that the clusters could have any (potentially non-convex) shape. For example, the means of two concentric circular clusters would coincide and lie outside of the actual clusters. Nevertheless, we could test DBSCAN pruning by taking a random subset of each cluster; however, in our early experiments, we observed that randomly selecting data points using our density-based criterion strongly underperformed compared to taking the most difficult examples.
> - **K-means runs on GPU using $faiss$ and is a lot faster than DBSCAN**: The time complexity of k-means clustering is $O(nki)$ (note that $k$ (num. clusters) and $i$ (num. iterations) << $n$) while DBSCAN can have a worst-case time complexity of $O(n^2)$. For example, based on our experiments, $k * i$ can be around 100,000 while $n$ can be >100,000,000 (1000 orders of magnitude higher).  We are uncertain how feasible DBSCAN would be for our large-scale datasets containing millions of training images. The $faiss$ implementation of k-means takes around 1-4 hours depending on the dataset size; a 1000-fold increase in clustering time would not be feasible for our dataset scales.
>
> ### Considering option (2):
> We could use DBSCAN to filter out outliers from the data before clustering. This might stabilize our kmeans algorithm, but would also introduce two new hyperparameters (the maximum distance between two points and the minimum number of samples in a cluster). An additional issue is the massively increased time complexity of the whole method (see the point above).

---

### Author Response · Authors · 2023-11-16
**General response to all reviewers.**

Dear reviewers,

We would like to thank you for taking the time to read and review our paper. We received very valuable feedback, incorporated it into the manuscript, and uploaded a new version. We include a detailed change log for our revised manuscript at the bottom of this post and respond to individual concerns in our individual responses.

In particular, all reviewers asked for results on more datasets and thus, we added new Tables (Tables 9 and 10) to our revised version with detailed results on the evaluation suite of DataComp which we also post here.

We also added more discussion and analysis of how training on pruned datasets affects model robustness.

### Change log:
- [all reviewers]: added Table 9 with detailed results on the full DataComp evaluation suite for models trained on DataComp.
- [all reviewers]: added Table 10 with detailed results on the full DataComp evaluation suite for models trained on LAION-CAT-440M as well as our filtered versions.
- [all reviewers]: To highlight the results on VTAB, ImageNet distribution shifts and retrieval tasks, we added a sentence to the paper abstract.
- [7m8Z]: added a paragraph on coreset selection to our related work section.
- [7m8Z]: reworded and restructured the methods section; moved details on the quadratic program to the Appendix.
- [mnGt, orZi]: better explained the non-triviality of scaling SSP Pruning to LAION / DataComp.
- [u5qD]: moved the QP program to the Appendix
- [u5qD, orZi]: rewrote parts of the results sections to improve clarity; added a discussion of how training on our pruned dataset influences robustness; added a new Figure on how longer training affects robustness.

---

> ### Author Response · Authors · 2023-11-16
> **Table 10: All validation datasets**
>
> Table: Results on 38 tasks from the DataComp evaluation suite when training on the filtered LAION-440M dataset. See also Tables 10 and 9 in the paper.
>
>
> |   | {}                     | Metric        | OpenCLIP | LAION-440M | SemDeDup | DBP    | SemD.+CS | DBP    | SSP    | DBP    | SSP    |
> |---|------------------------|---------------|----------|------------|----------|--------|----------|--------|--------|--------|--------|
> |   | Datset Size            |               | 400M     | 440M       | 277M     | 222M   | 222M     | 166M   | 166M   | 112M   | 112M   |
> |   | Num. sam. seen         |               | 12.8B    | 12.7B      | 8.8B     | 7.1B   | 7.1B     | 5.3B   | 5.3B   | 3.6B   | 3.6B   |
> |   | Tr. Cost \%            |               | 100\%    | 99.2\%     | 69.3\%   | 55.4\% | 55.4\%   | 41.6\% | 41.6\% | 27.7\% | 27.7\% |
> |   | ImageNet 1k            | Acc           | 62.93    | 64.07      | 64.32    | 65.17  | 61.64    | 65.46  | 65.09  | 64.09  | 62.77  |
> |   | ImageNet Sketch        | Acc           | 49.38    | 49.78      | 49.88    | 49.5   | 47.2     | 49.21  | 49.18  | 47.36  | 46.76  |
> |   | ImageNet v2            | Acc           | 55.06    | 55.89      | 56.11    | 56.77  | 53.14    | 57.62  | 56.79  | 56.0   | 54.94  |
> |   | ImageNet-A             | Acc           | 21.72    | 25.04      | 25.65    | 27.23  | 23.48    | 26.92  | 26.25  | 25.83  | 22.71  |
> |   | ImageNet-O             | Acc           | 53.45    | 50.6       | 51.85    | 52.5   | 52.7     | 53.05  | 55.25  | 54.6   | 55.0   |
> |   | ImageNet-R             | Acc           | 73.42    | 72.25      | 73.0     | 72.09  | 69.9     | 71.33  | 71.81  | 68.77  | 67.27  |
> |   | ObjectNet              | Acc           | 43.87    | 47.1       | 47.5     | 49.3   | 45.46    | 47.53  | 47.6   | 46.02  | 43.67  |
> |   | Caltech-101            | Acc           | 91.18    | 90.38      | 89.97    | 90.28  | 89.01    | 90.21  | 90.81  | 89.91  | 89.74  |
> |   | CIFAR-100              | Acc           | 70.29    | 76.48      | 75.47    | 75.33  | 73.88    | 75.97  | 76.03  | 75.31  | 73.37  |
> |   | CLEVR Counts           | Acc           | 16.24    | 23.57      | 17.21    | 33.23  | 22.7     | 25.44  | 18.37  | 21.49  | 19.97  |
> |   | CLEVR Distance         | Acc           | 23.91    | 14.97      | 17.88    | 24.51  | 22.59    | 20.45  | 18.12  | 23.77  | 24.48  |
> |   | Describable Textures   | Acc           | 54.57    | 54.2       | 54.79    | 56.06  | 48.46    | 53.94  | 52.61  | 46.17  | 46.44  |
> |   | EuroSAT                | Acc           | 51.43    | 52.72      | 44.37    | 55.76  | 47.07    | 59.56  | 47.65  | 44.69  | 41.15  |
> |   | KITTI Vehicle Distance | Acc           | 28.97    | 13.92      | 9.56     | 17.02  | 14.06    | 25.74  | 10.97  | 23.77  | 23.49  |
> |   | Oxford Flowers-102     | Acc           | 66.18    | 62.66      | 63.54    | 64.59  | 60.98    | 65.84  | 65.71  | 67.47  | 68.91  |

---

> ### Author Response · Authors · 2023-11-16
> **Table Cont.**
>
> |   | {}                     | Metric        | OpenCLIP | LAION-440M | SemDeDup | DBP    | SemD.+CS | DBP    | SSP    | DBP    | SSP    |
> |---|------------------------|---------------|----------|------------|----------|--------|----------|--------|--------|--------|--------|
> |   | Datset Size            |               | 400M     | 440M       | 277M     | 222M   | 222M     | 166M   | 166M   | 112M   | 112M   |
> |   | Num. sam. seen         |               | 12.8B    | 12.7B      | 8.8B     | 7.1B   | 7.1B     | 5.3B   | 5.3B   | 3.6B   | 3.6B   |
> |   | Tr. Cost \%            |               | 100\%    | 99.2\%     | 69.3\%   | 55.4\% | 55.4\%   | 41.6\% | 41.6\% | 27.7\% | 27.7\% |
> |   | Oxford-IIIT Pet        | Acc           | 86.71    | 87.33      | 88.56    | 88.18  | 84.74    | 88.36  | 88.9   | 87.97  | 87.51  |
> |   | PatchCamelyon          | Acc           | 55.91    | 58.69      | 55.27    | 49.98  | 49.86    | 49.04  | 49.89  | 49.53  | 49.97  |
> |   | RESISC45               | Acc           | 54.54    | 58.51      | 59.27    | 58.35  | 56.6     | 59.52  | 52.3   | 52.49  | 45.06  |
> |   | SVHN                   | Acc           | 30.39    | 28.28      | 29.49    | 22.85  | 24.36    | 12.74  | 11.84  | 16.84  | 7.05   |
> |   | SUN397                 | Acc           | 66.99    | 66.87      | 66.29    | 66.55  | 65.21    | 67.03  | 66.86  | 64.36  | 63.52  |
> |   | Flickr                 | Recall        | 70.21    | 76.04      | 76.37    | 76.74  | 73.86    | 75.25  | 75.21  | 73.15  | 72.81  |
> |   | MSCOCO                 | Recall        | 43.93    | 48.06      | 48.86    | 48.44  | 45.76    | 48.38  | 46.98  | 45.65  | 44.24  |
> |   | WinoGAViL              | Jaccard Score | 40.8     | 44.91      | 42.58    | 42.92  | 41.72    | 38.93  | 37.8   | 37.26  | 36.81  |
> |   | CIFAR-10               | Acc           | 90.74    | 93.75      | 93.79    | 93.82  | 92.93    | 93.28  | 93.68  | 92.31  | 92.41  |
> |   | Country211             | Acc           | 14.75    | 14.81      | 14.78    | 15.64  | 13.8     | 14.75  | 14.0   | 13.78  | 12.68  |
> |   | FGVC Aircraft          | Acc           | 16.58    | 12.42      | 13.79    | 14.47  | 11.34    | 14.01  | 13.24  | 17.21  | 11.85  |
> |   | Food-101               | Acc           | 80.86    | 81.29      | 81.46    | 82.41  | 79.13    | 82.55  | 80.2   | 79.98  | 75.25  |
> |   | GTSRB                  | Acc           | 41.99    | 35.61      | 44.98    | 30.58  | 38.58    | 24.11  | 31.88  | 20.17  | 20.86  |
> |   | MNIST                  | Acc           | 37.33    | 37.23      | 34.03    | 23.79  | 29.22    | 16.63  | 14.49  | 9.17   | 11.09  |
> |   | Pascal VOC 2007        | Acc           | 75.82    | 79.51      | 79.77    | 80.37  | 77.96    | 79.17  | 78.88  | 78.72  | 78.59  |
> |   | Rendered SST2          | Acc           | 52.28    | 49.53      | 49.37    | 48.16  | 52.28    | 50.25  | 48.54  | 49.97  | 47.17  |
> |   | Stanford Cars          | Acc           | 79.26    | 74.65      | 75.41    | 74.39  | 73.81    | 66.35  | 66.35  | 55.11  | 60.71  |
> |   | STL-10                 | Acc           | 95.6     | 95.73      | 96.9     | 96.3   | 96.39    | 96.28  | 96.14  | 95.71  | 95.36  |
> |   | iWildCam               | Acc           | 7.44     | 8.08       | 7.28     | 7.86   | 7.27     | 8.82   | 7.57   | 7.17   | 5.0    |
> |   | Camelyon17             | Acc           | 47.04    | 50.2       | 62.71    | 49.84  | 50.0     | 48.91  | 50.1   | 49.6   | 49.84  |
> |   | FMoW                   | Acc           | 12.96    | 12.38      | 14.75    | 13.96  | 14.12    | 0.0    | 8.87   | 0.0    | 0.0    |
> |   | Dollar Street          | Acc           | 54.91    | 58.41      | 56.43    | 57.94  | 55.84    | 55.37  | 55.96  | 57.48  | 55.84  |
> |   | GeoDE                  | Acc           | 83.8     | 86.01      | 84.76    | 84.72  | 84.01    | 84.22  | 84.76  | 84.72  | 83.76  |
> |   | Average                |               | 52.72    | 52.95      | 53.11    | 53.09  | 51.34    | 51.64  | 50.7   | 49.83  | 48.63  |

---

### Meta-Review · Area_Chair_UZwD · 2023-12-07

**Metareview:**

A) This paper presents a pruning method for reducing size of datasets while improving/maintaining accuracy. The paper scales the pruning approach to DataComp and LAION datasets and achieve strong accuracies after pruning the dtaset
B) The reviewers agree that the paper has thorough experiments, is reasonably well written and tackles an important problem. Reviewer u5qD summarizes the strengths well: "The dimensions of ablations, and the high level takeaways from those ablations in the results section are clear -- DBP outperforms SSP Pruning, outperforms CLIP filtering, DBP outperforms SemDeDup alone, and Deduplicaiton is crucial to the success of the method. These takeaways make the contribution quite compelling."
C) Unfortunately despite this there is a lack of cohesion and strength in results when compared to other methods especially on tasks other than ImageNet.  For example "DBP does a tiny bit better than TMARS on Imagenet, but more significantly worse on Imagenet Dist and Average but a discussion of the latter point is entirely missing". Overall the reviewers seem to agree that the results section seem a bit scattershot and confusing especially when compared to other work


Despite this I think the paper is experimentally solid and tackles and important problem, and I believe the authors have addressed almost all points brought up by the reviewers in rebuttal, specifically the problems with evaluation and the issue of using CLIP filtering as a baseline (brought up by reviewer mnGt).

 For this reason I vote to accept this paper.

**Justification For Why Not Higher Score:**

The paper does have some issues with the strength of results, and comparison to more recently published filtering baselines, while this cannot be held against the paper for acceptance it does prevent it from getting a higher score in my book.

**Justification For Why Not Lower Score:**

The paper is well written has *very* extensive experiments and shows how the proposed approach does perform better than similar approaches for large scale datasets. I think this type of good experimentation should be encouraged for ICLR and thus deserves acceptance.

---

### Decision · Program_Chairs · 2024-01-16

Accept (poster)